# Small Molecules with Thiourea Skeleton Induce Ethylene Response in *Arabidopsis*

**DOI:** 10.3390/ijms241512420

**Published:** 2023-08-04

**Authors:** Tomoyuki Koyama, Honami Zaizen, Ikuo Takahashi, Hidemitsu Nakamura, Masatoshi Nakajima, Tadao Asami

**Affiliations:** Graduate School of Agricultural and Life Sciences, The University of Tokyo, Tokyo 113-8657, Japan; tomoyukikoyama800@g.ecc.u-tokyo.ac.jp (T.K.); takahashi.190@gmail.com (I.T.); hidemitsunakamura87@gmail.com (H.N.); nkjmmsts@gmail.com (M.N.)

**Keywords:** plant hormone, ethylene mimic, triple response, thiourea

## Abstract

Ethylene is the only gaseous plant hormone that regulates several aspects of plant growth, from seedling morphogenesis to fruit ripening and organ senescence. Ethylene also stimulates the germination of *Striga hermonthica*, a root parasitic weed that severely damages crops in sub-Saharan Africa. Thus, ethylene response stimulants can be used as weed and crop control agents. Ethylene and ethephon, an ethylene-releasing compound, are currently used as ethylene response inducers. However, since ethylene is a gas, which limits its practical application, we targeted the development of a solid ethylene response inducer that could overcome this disadvantage. We performed chemical screening using *Arabidopsis thaliana* “triple response” as an indicator of ethylene response. After screening, we selected a compound with a thiourea skeleton and named it ZKT1. We then synthesized various derivatives of ZKT1 and evaluated their ethylene-like activities in *Arabidopsis*. Some derivatives showed considerably higher activity than ZKT1, and their activity was comparable to that of 1-aminocyclopropane-1-carboxylate. Mode of action analysis using chemical inhibitors and ethylene signaling mutants revealed that ZKT1 derivatives activate the ethylene signaling pathway through interactions with its upstream components. These thiourea derivatives can potentially be potent crop-controlling chemicals.

## 1. Introduction

Ethylene (C_2_H_4_) is a gaseous plant hormone that regulates several plant growth processes. For example, ethylene regulates the morphogenesis of seedlings and promotes the senescence of leaf and flower petals, fruit ripening, and organ abscission. Ethylene is also involved in plant responses to biotic and abiotic stresses [1,2].

In higher plants, ethylene is synthesized from methionine [3]. Methionine is transformed into *S*-adenosylmethionine (SAM) by SAM synthase and then converted to 1-aminocyclopropane-1-carboxylate (ACC) by ACC synthase [4,5]. ACC is oxidized by ACC oxidase (ACO) and converted to ethylene [6,7]. The ethylene signaling pathway has been revealed through genetic analysis using ethylene-related mutants. Ethylene is perceived by ER membrane-localized ethylene receptors, which resemble bacterial histidine kinases [8,9,10]. In the absence of ethylene, ethylene receptors activate a Raf-like kinase CONSTITUTIVE TRIPLE RESPONSE1 (CTR1), and CTR1 represses ETHYLENE INSENSITIVE2 (EIN2), which is a Nramp homologous positive regulator of ethylene signaling, through the phosphorylation of the C-terminal domain of EIN2 (EIN2-C) [11,12,13,14,15]. In contrast, when ethylene binds to ethylene receptors, the ethylene receptors, and therefore CTR1, are inactivated, leading to the decreased phosphorylation of EIN2-C. Dephosphorylated EIN2-C is cleaved and indirectly activates the master transcription factors involved in ethylene signaling, including ETHYLENE INSENSITIVE3 (EIN3) and EIN3-like1 (EIL1) [15,16,17,18]. Finally, EIN3 and EIL1 promote the expression of ethylene-responsive genes and the ethylene response [19].

Because ethylene is involved in plant processes of agronomic and horticultural significance, efforts have been made to develop chemicals to control ethylene functions. These efforts have targeted non-gaseous chemicals because the gaseous nature of ethylene makes its application in open fields problematic [20]. A commercially available solid compound, 2-chloroethylphosphonic acid (ethephon), releases ethylene upon treatment to plant tissues [21]. Ethephon has been registered as a pesticide for several crops and is used to promote fruit ripening, abscission, and induction of flowering [22]. Ethylene can also be utilized to control *Striga hermonthica*, a root parasitic weed that causes severe damage to crops in sub-Saharan Africa [23]. Because *S. hermonthica* is an obligate parasite and its germination is promoted by ethylene, ethylene can be used to control this weed using a strategy termed suicidal germination, which utilizes germination stimulants to induce *S. hermonthica* germination in the absence of host plants [24]. Furthermore, ethylene fumigation has successfully eradicated *Striga asiatica* in North Carolina, United States [25]. However, costly equipment is necessary for ethylene fumigation, and this method is not an optimal solution for sub-Saharan Africa [20]. Therefore, solid ethylene-mimics with affordable costs are anticipated in the field of agriculture.

To explore solid-type ethylene mimics, we screened a chemical library using the *Arabidopsis* triple response assay [26]. We then conducted structure-activity relationship research and selected compounds with enhanced ethylene-like activity. Finally, chemical biological, and genetic studies were conducted to elucidate the mechanisms of action of the selected compounds.

## 2. Results

### 2.1. Screening of Chemicals That Stimulate the Triple Response in Arabidopsis Thaliana from a Chemical Library

First, we screened compounds with ethylene-like activity from 2500 compounds involved in a library of synthetic chemicals supplied by Maybridge. To evaluate the ethylene-like activity of compounds, we used the “triple response” of *A. thaliana.* Sterilized *A. thaliana* seeds were sown on 1/2 MS agar medium containing 20 μM of each chemical, grown for 3 d under dark conditions, and the morphology of seedlings was observed. One compound with a thiourea skeleton (Figure 1a) induced the “triple response” of *A. thaliana*. Therefore, we selected this compound as the lead compound for use as an ethylene mimic and named it ZKT1.

### 2.2. Synthesis of Derivatives of ZKT1

Synthetic derivatives of ZKT1 were prepared to improve its ethylene-like activity. ZKT1 derivatives were designed by replacing the norbornane moiety with a substituted phenyl moiety. ZKT1 derivatives were synthesized via a single-step reaction between primary amines and isothiocyanates in acetone, dimethylformamide, or dichloromethane (Figure 1b). The 17 synthesized ZKT1 derivatives are listed in Table 1 and their nuclear magnetic resonance (NMR) spectra are shown in Appendix A.

### 2.3. Evaluation of Ethylene-Like Activity of ZKT1 Derivatives

Dark-grown seedlings of *A. thaliana* show the triple response to treatment with ethylene, or its precursor, ACC. The triple response of *A. thaliana* consists of inhibition of hypocotyl and root elongation, promotion of radical swelling of the hypocotyl, and exaggeration of apical hook bending. Of these three traits, we used the inhibition of hypocotyl elongation and the exaggeration of apical hook bending as indicators of the triple response to evaluate the ethylene-like activity of ZKT1 derivatives. To evaluate the exaggeration of apical hook bending, we measured the angles of the apical hooks using the methods introduced previously [27]. *A. thaliana* seedlings were grown under dark conditions for 3 d after treatment with 5 μM of ACC or 20 μM of ZKT1 derivatives, and hypocotyl length and angle of curvature of hooks were measured (Figure 2a,b). Because seedlings treated with ZKT9 had open cotyledons, the apical hook angles of the ZKT9-treated seedlings could not be defined. The hypocotyl length of mock treatment was 8.23 ± 0.28 (mm) and was reduced to 3.23 ± 0.11 (mm) by 5 μM ACC treatment. The apical hook angle of mock treatment was 189.2 ± 16.5 (degrees) and was increased to 264.4 ± 9.1 (degrees) by 5 μM of ACC treatment. These results demonstrate the success of our experimental method. Upon 20 μM ZKT1 treatment, the hypocotyl length was reduced to 4.46 ± 0.19 (mm), and the apical hook angle was increased to 235.7 ± 7.8 (degrees). All ZKT1 derivatives, excluding ZKT7 and ZKT9, induced an ethylene response: the inhibition of hypocotyl elongation and exaggeration of apical hook bending. ZKT7, which had a urea skeleton instead of a thiourea skeleton, inhibited hypocotyl elongation but did not induce exaggeration of apical hook bending. ZKT9, which had chlorine at the 4-position of the phenyl moiety, exhibited significantly stronger inhibition of hypocotyl elongation than ACC and induced cotyledon opening. ZKT2, 3, 4, 11, and 18 inhibited hypocotyl elongation more strongly than ZKT1 and increased the apical hook angle to the same extent as 5 μM of ACC. Therefore, we compared the activities of ZKT2, 3, 4, 11, and 18 at 10 μM with those of ACC at 5 μM (Figure 2c,d). At 10 μM, these five ZKT compounds inhibited hypocotyl elongation to the same extent as that of ACC at 5 μM, and the apical hook angles of seedlings treated with these five ZKT compounds were comparable to those treated with ACC.

The concentration dependency of ethylene-like activity of ZKT2 and 18 were subsequently evaluated because they demonstrated higher potency than other compounds in both tests at concentrations of 20 and 10 μM. *A. thaliana* seedlings were grown for 3 d with 0.001 to 100 μM of ACC or 0.001 to 20 μM of ZKT2 or ZKT18, and the results are shown for hypocotyl length and apical hook angle (Figure 2e) with EC_50_ values (Table 2). The EC_50_ value calculated from the hypocotyl length of the ACC was approximately one-eighth and one-tenth that of ZKT2 and ZKT18, respectively. However, the slopes of the dose-response curves of ZKT2 and ZKT18 were substantially steeper than those of ACC, indicating that the concentrations at which the compounds reached their maximum effects differed by a factor of only two to four.

### 2.4. Analysis of the Mechanism of Action of ZKT Compounds

The mechanisms of action of ZKT2 and 18 were analyzed using chemical inhibitors and ethylene signaling mutants. First, dark-grown seedlings of *A. thaliana* were treated with ZKT2 or ZKT18 in the presence or absence of α-aminoisobutyric acid (AIBA), an ACO inhibitor [28]. This analysis aimed to determine whether ZKT compounds act upstream in the ethylene biosynthesis pathway. The hypocotyl length and apical hook angle of dark-grown seedlings of *A. thaliana* in these tests are shown in Figure 3a,b. AIBA treatment significantly recovered the hypocotyl length of seedlings treated with 5 μM of ACC. In addition, AIBA treatment reduced the apical hook angle of seedlings treated with 5 μM of ACC significantly. In contrast, the hypocotyl length and apical hook angle of seedlings treated with 10 μM of ZKT2 or ZKT18 with or without AIBA treatment were approximately the same. In addition, AIBA did not reduce the apical hook angle of seedlings treated with 10 μM of ZKT2. Although AIBA slightly reduced the apical hook angle of seedlings treated with 10 μM of ZKT18, the extent of reduction was substantially smaller than that of ACC. These results indicate that ZKT1 derivatives act downstream of the ethylene biosynthesis pathway.

Subsequently, the dark-grown seedlings of the *ein2-5* mutant were treated with ZKT2 or ZKT18 (Figure 3c,d). ACC did not inhibit hypocotyl elongation or induce exaggeration of apical hook curvature in the *ein2-5* mutant. ZKT2 inhibited hypocotyl elongation and induced the exaggeration of apical hook curvature in the *ein2-5* mutant; however, the extent of inhibition or induction was substantially less than that in Col-0 seedlings. In contrast, ZKT18 did not inhibit hypocotyl elongation or exaggerate apical hook bending in the *ein2-5* mutant. These results suggest that ZKT compounds activate the ethylene signaling pathway by interacting with ethylene-signaling components upstream of EIN2.

Third, to further confirm that the ZKT compounds activated ethylene signaling, dark-grown seedlings of the *ein3 eil1* double mutant were treated with ZKT2 or ZKT18 (Figure 3e,f). ACC, ZKT2, and ZKT18 did not inhibit hypocotyl elongation in the *ein3 eil1* double mutant. In addition, ACC and ZKT18 did not induce exaggeration of the apical hook curvature. ZKT2 slightly increased the apical hook angle; however, the difference was not significant. Therefore, these ZKT1 derivatives were confirmed to activate the ethylene signaling pathway.

Finally, fluctuations in the expression of *ERF1*, an ethylene-responsive gene, were measured after treatment with ACC and ZKT2. The expression levels of *ERF1* was investigated by RT-qPCR at 3 and 24 h after treatment with 10 μM of ACC or 30 μM of ZKT2 in 7 d old Col-0 seedlings (Figure 3g). Three hours after ACC treatment, a significant increase in *ERF1* expression was noted. At 24 h after ACC treatment, the expression level of *ERF1* remained high but was slightly lower than that observed at 3 h after treatment. ZKT2 also induced an increase in the expression level of *ERF1* at both 3 and 24 h after treatment. In contrast to ACC, the expression of *ERF1* continued to increase until 24 h after ZKT2 treatment.

## 3. Discussion

### 3.1. Structure-Activity Relationship of ZKT1 Derivatives

In this study, we screened small chemicals that induce the “triple response” in *A. thaliana* and selected the thiourea derivative ZKT1 as a compound for further potentiation. Through structure development based on ZKT1, we successfully designed chemicals with enhanced biological activity for inducing the “triple response” in *A. thaliana* etiolated seedlings. Considering that the urea derivative ZKT7 did not show ethylene-like activity, it can be inferred that the sulfur atom in the thiourea skeleton is essential for the biological activity of the ZKT1 derivatives. Among the synthesized derivatives of ZKT1, compounds ZKT2, 3, 4, 11, and 18 exhibited the strongest activity. In contrast, compounds ZKT6, 8, 10, 14, and 16 had reduced biological activity. Furthermore, 1-phenyl-3-(2-(pyridin-2-yl)ethyl)thiourea derivatives bearing substituents at the 4-position of the phenyl moiety (ZKT2, 3 and 4) generally exhibited high activity. Among compounds with varying carbon chain lengths attached to the pyridine ring, 1-phenyl-3-(pyridin-2-yl)thiourea (ZKT5) exhibited moderately high activity, whereas 1-phenyl-3-(pyridin-2-ylmethyl)thiourea (ZKT6) showed moderately low activity. Furthermore, the derivatives of ZKT6 with a single substituent at the 4-position of the phenyl moiety (ZKT9, 10, 14, and 15) exhibited weak biological activity, excluding ZKT15. Among the ZKT6 derivatives with two chlorine atoms on the phenyl moiety, ZKT11, which had chlorine atoms at the 3- and 4-positions, showed the highest activity. Among the ZKT1 derivatives with a phenyl moiety instead of a pyridinyl moiety (ZKT16, 17, and 18), 1-phenetyl-3-phenylthiourea (ZKT18) exhibited the highest activity. Therefore, the nitrogen atom in the pyridinyl moiety is unnecessary for the ethylene-like activity of the thiourea derivatives. Notably, ZKT17, in which the pyridinyl moiety of ZKT6 was replaced with a phenyl moiety, showed increased activity. However, ZKT16, in which the pyridinyl moiety of ZKT5 was replaced with a phenyl moiety, exhibited decreased activity. This result suggests that the nitrogen atom in the pyridinyl moiety may influence the biological activity of ZKT1 derivatives, although it is not essential for their activity. Based on the EC_50_ values, ACC exhibited approximately eight–ten times stronger activity than ZKT2 and ZKT18. However, the slope of the dose-response curve for ACC was substantially gentler than those for ZKT2 and ZKT18, resulting in a difference of only two–four times the concentration required for response saturation. Therefore, the difference in the amount required to use ZKT compounds as ethylene response inducers was not as significant as the difference in the EC_50_ values compared with ACC.

### 3.2. Possible Mechanisms for ZKT1 Derivatives to Express Ethylene-Like Activity

Based on the analysis of the biological activities of ZKT2 and ZKT18 using a chemical inhibitor and ethylene-insensitive mutants, we concluded that the ZKT1 derivatives exhibited ethylene-like activity through activation of the ethylene signaling pathway. Since the inhibitor of ACC oxidase did not inhibit the ethylene response induced by ZKT2 and ZKT18, ZKT1 derivatives were confirmed to not enhance ethylene biosynthesis but rather stimulate the ethylene signaling pathway. ZKT18 did not induce an ethylene response in the ethylene-insensitive mutants *ein2-5* and *ein3 eil1*, indicating that ZKT1 derivatives stimulate the ethylene signaling pathway by affecting upstream signaling components, such as ethylene receptors or CTR1. In contrast, ZKT2 suppressed hypocotyl elongation in the *ein2-5* mutant and slightly increased the apical hook angle in both the *ein2-5* and *ein3 eil1* mutants. Recent studies have suggested that ethylene signals can be transduced from ethylene receptors independent of CTR1 and EIN2. For example, cytokinin signaling components, such as Arabidopsis histidine-containing phosphotransfer proteins and Arabidopsis response regulators, have been proposed to play a role in ethylene signaling [29,30,31]. ZKT2 may stimulate these alternative signaling pathways, bypassing the traditional ethylene signaling pathway.

Copper cofactors in the +1-oxidation state are essential for the binding of ethylene to ethylene receptors and for receptor function [32,33,34]. Monovalent copper atoms are primarily delivered to ethylene receptors by stepwise transfer from ANTIOXIDANT-1 or COPPER TRANSPORT PROTEIN to RESPONSE-TO-ANTAGONIST-1 (RAN1), followed by the transfer of copper from RAN1 to ethylene receptors [35]. Seedlings of *ran1-3*, a strong loss-of-function mutant of RAN1, exhibit a constitutive triple-response phenotype [36]. Thiourea forms a complex with monovalent copper atoms, and thiourea ligands are coordinated with the monovalent copper atoms through sulfur atoms [37,38]. In addition, 1-phenyl-3-(pyridine-2-yl)thiourea (ZKT5 in our study) and a copper (I) cation have been reported to form a coordination polymer and dinuclear complex [39]. In this case and in former studies, the ligands are coordinated to a monovalent copper atom through sulfur atoms, and no coordination between nitrogen atoms of pyridine rings and a copper center has been observed. These facts may explain the reason for the nitrogen atom of the pyridine ring being not essential for the ethylene-like activity of ZKT1 derivatives. In addition, thiourea derivatives have been reported to coordinate copper (II) cations via sulfur atoms [40]. Considering these facts, we estimated that ZKT1 derivatives exhibit ethylene-like activity through coordination with copper cations in the plant body, and we propose two putative mechanisms for the ethylene-like activity of ZKT1 derivatives. First, ZKT1 derivatives bind to copper cofactors in ethylene receptors and activate ethylene signaling. Second, ZKT1 derivatives either deprive the copper cofactor of ethylene receptors or reduce the concentration of available copper atoms, leading to conditions similar to those in the *ran1-3* mutant. The elucidation of the mode of action of ZKT1 derivatives is the remaining question of this research.

### 3.3. Comparison of ZKT1 Derivatives with Existing Ethylene Mimics

To date, three compounds–neocuproine, EH-1, and triplin have been reported to act as ethylene response stimulants. Neocuproine is a copper chelator and treatment with 100 μM induced an ethylene response in *Arabidopsis* etiolated seedlings [41]. However, the application of neocuproine is difficult because of its cytotoxicity [42]. EH-1 and its derivatives have pyrazole ring and sulfonamide moiety in common, and treatment with 10 μM of EH-1 induced the “triple response” of *Arabidopsis* [22]. Triplin is a compound with a thiourea moiety, which is common in ZKT1 derivatives, and exhibits ethylene-like activity at concentrations higher than 20 μM [43]. ZKT2 and 18, the ZKT1 derivatives with the highest activity, induced an ethylene response at concentrations higher than 5 μM, and their activity was saturated when the concentration was higher than 10 μM (Figure 2e). In summary, ZKT1 derivatives generally showed stronger activity than existing ethylene mimics. EH-1 has activity comparable to that of the strongest ZKT1 derivatives but requires 12 h for synthesis and purification by column chromatography, which makes large-scale preparation difficult. In contrast, ZKT1 derivatives can be synthesized in 3 h and can easily be purified by recrystallization. This preparation convenience encourages the application of ZKT1 derivatives in agriculture.

## 4. Materials and Methods

### 4.1. General

The chemicals used for the synthesis were obtained from Kanto Chemicals Co. Ltd. (Tokyo, Japan), Tokyo Kasei Co. Ltd. (Tokyo, Japan), FUJIFILM Wako Pure Chemical Corporation (Osaka, Japan), Nacalai Tesque Co. Ltd. (Kyoto, Japan), and Sigma-Aldrich Japan LLC (Tokyo, Japan). All reagents used were of the highest grade available. ^1^H NMR spectra were recorded on a 500 MHz JEOL ECA 500 II. Chemical shifts of ^1^H NMR were reported in ppm relative to internal tetramethylsilane (internal standard, 0.0 ppm), with coupling constants (*J*) given in Hz. Chemical shifts of ^13^C NMR were recorded with the same machine operating at 125 MHz with complete proton decoupling (internal standard CDCl_3_: 77.0 ppm, DMSO-d_6_: 39.5 ppm), with coupling constants (*J*) given in Hz when the compounds had fluorine atoms. All materials were used as commercially supplied. All reactions were monitored by thin-layer chromatography using Merck silica gel 60F254. Flash column chromatography was performed using a Wakogel C-300HG column. High-resolution mass spectra were obtained by electrospray ionization coupled with a time-of-flight analyzer (Triple TOF 5600+ system, AB Sciex LLC, Framingham, MA, USA).

### 4.2. Chemical Synthesis

1-(4-*tert*-butylphenyl)-3-(2-(pyridin-2-yl)ethyl)thiourea (ZKT2)

2-(2-Aminoethyl)pyridine (237 μL, 2.00 mmol) was added to the solution of 4-*tert*-butylphenyl isothiocyanate (383 mg, 2.00 mmol) in 23 mL dichloromethane, and was stirred for an hour. The reaction mixture was evaporated and passed through a short silica gel column eluted by ethyl acetate. The resulting residue was purified by recrystallization from ethyl acetate and hexane to obtain 505 mg of white plate crystal (yield: 80.5%). ^1^H-NMR (CDCl_3_) δ 8.13 (d, *J* = 4.0 Hz, 1H), 7.83 (br, 1H), 7.59 (td, *J* = 7.5 Hz, 2.0 Hz, 1H), 7.51 (br, 1H), 7.40 (dt, *J* = 9.3 Hz, 2.3 Hz, 2H), 7.15 (d, *J* = 8.0 Hz, 1H), 7.06–7.10 (m, 3H), 4.05 (q, *J* = 5.8 Hz, 2H), 3.06 (t, *J* = 6.0 Hz, 2H), 1.35 (s, 9H). ^13^C-NMR (CDCl_3_) δ 179.9, 159.6, 150.0, 148.5, 136.7, 133.4, 126.6, 125.1, 123.5, 121.4, 44.4, 35.6, 34.6, 31.3. HRMS (ESI): *m/z* [M + H]^+^ calcd for C_18_H_24_N_3_S 314.1685, found 314.1681.

1-(4-bromophenyl)-3-(2-(pyridin-2-yl)ethyl)thiourea (ZKT3)

2-(2-Aminoethyl)pyridine (35.6 μL, 0.300 mmol) was added to the solution of 4-bromophenyl isothiocyanate (64.2 mg, 0.300 mmol) in 2.4 mL acetone, and was refluxed for 3 h. The reaction mixture was evaporated and purified by recrystallization from ethanol to obtain 52.9 mg of pale yellow solid (yield: 52.4%). ^1^H-NMR (CDCl_3_) δ 8.17 (br, 2H), 7.80 (br, 1H), 7.61 (td, *J* = 7.8 Hz, 1.8 Hz, 1H), 7.52 (dt, *J* = 9.2 Hz, 2.6 Hz, 2H), 7.06–7.17 (m, 4H), 4.04 (q, *J* = 5.5 Hz, 2H), 3.06 (t, *J* = 5.5 Hz, 2H). ^13^C-NMR (CDCl_3_) δ 179.8, 159.6, 148.5, 136.9, 135.4, 132.8, 126.8, 123.6, 121.7, 120.1, 44.7, 35.3. HRMS (ESI): *m/z* [M + H]^+^ calcd for C_14_H_15_BrN_3_S 336.0165, found 336.0159.

1-(2-(pyridin-2-yl)ethyl)-3-(4-(trifluoromethyl)phenyl)thiourea (ZKT4)

ZKT4 was synthesized from 4-(trifluoromethyl)phenyl isothiocyanate (63.0 mg, 0.307 mmol) and 2-(2-aminoethyl)pyridine (36.5 μL, 0.307 mmol) by the same method as ZKT3, and 69.7 mg of pale yellow solid was obtained (yield: 69.7%). ^1^H-NMR (CDCl_3_) δ 8.48 (br, 1H), 8.13 (br, 1H), 7.80 (br, 1H), 7.62–7.67 (m, 3H), 7.31 (br, 2H), 7.18 (d, *J* = 7.5 Hz, 1H), 7.14 (t, *J* = 6.0 Hz, 1H), 4.08 (q, *J* = 5.3 Hz, 2H), 3.09 (t, *J* = 5.8 Hz, 2H). ^13^C-NMR (CDCl_3_) δ 179.4, 159.6, 148.4, 139.9, 137.0, 127.8 (q, *J* = 33.0 Hz), 126.8, 124.0, 123.8 (q, *J* = 270.2 Hz), 123.6, 121.8, 44.7, 35.2. HRMS (ESI): *m/z* [M + H]^+^ calcd for C_15_H_15_F_3_N_3_S 326.0933, found 326.0920.

1-phenyl-3-(pyridin-2-yl)thiourea (ZKT5)

2-Aminopyridine (41.0 mg, 0.436 mmol) was added to the solution of phenyl isothiocyanate (51.7 μL, 0.436 mmol) in 3.3 mL acetone, and was refluxed for 3 h. The precipitated solid was filtered and washed with hexane to obtain 60.7 mg of white solid (yield: 60.7%). ^1^H-NMR (CDCl_3_) δ 13.66 (s, 1H), 8.82 (s, 1H), 8.22–8.24 (m, 1H), 7.67–7.71 (m, 3H), 7.41–7.44 (m, 2H), 7.25–7.28 (m, 1H), 7.00–7.03 (m, 1H), 6.85 (d, *J* = 8.0 Hz, 1H). ^13^C-NMR (CDCl_3_) δ 178.8, 153.2, 145.6, 139.0, 138.6, 128.7, 126.3, 125.0, 118.3, 112.5. HRMS (ESI): *m/z* [M + H]^+^ calcd for C_12_H_12_N_3_S 230.0746, found 230.0746.

1-phenyl-3-(pyridin-2-ylmethyl)thiourea (ZKT6)

Phenyl isothiocyanate (237 μL, 2.00 mmol) was added to the solution of 2-(2-aminomethyl)pyridine (202 μL, 2.00 mmol) in 23 mL dichloromethane, and the reaction mixture was stirred at room temperature for an hour. The reaction mixture was evaporated and passed through a short silica gel column eluted by ethyl acetate. The resulting residue was further purified by recrystallization from ethyl acetate and hexane to obtain 431 mg of white powder (yield: 88.7%). ^1^H-NMR (CDCl_3_) δ 8.42 (d, *J* = 9.5 Hz, 1H), 7.87 (br, 1H), 7.80 (br, 1H), 7.67 (td, *J* = 7.6 Hz, 1.7 Hz, 1H), 7.43–7.46 (m, 2H), 7.28–7.31 (m, 4H), 7.17–7.19 (m,1H), 4.94 (s, 2H). ^13^C-NMR (CDCl_3_) δ 179.9, 155.3, 148.7, 136.8, 136.4, 129.9, 126.7, 124.6, 122.4, 122.0, 49.8. HRMS (ESI): *m/z* [M + H]^+^ calcd for C_13_H_14_N_3_S 244.0903, found 244.0905.

1-phenyl-3-(2-(pyridin-2-yl)ethyl)urea (ZKT7)

2-(2-Aminoethyl)pyridine (49.1 μL, 0.414 mmol) was suspended in 1.0 mL hexane and was added to phenyl isocyanate (44.9 μL, 0.414 mmol) solution in 1.0 mL hexane at ice temperature. Then, the ice bath was removed and the reaction mixture was stirred at room temperature for 3 h. The precipitated solid was filtered and washed with hexane to obtain 46.3 mg pale yellow solid (yield: 46.3%). ^1^H-NMR (CDCl_3_) δ 8.44–8.45 (m, 1H), 7.59–7.63 (m, 1H), 7.25–7.30 (m, 4H), 7.18–7.19 (m, 1H), 7.13–7.15 (m, 1H), 7.05–7.09 (m, 1H), 6.68–6.79 (m, 1H), 5.80 (br, 1H), 3.68 (q, *J* = 6.0 Hz, 2H), 3.02 (t, *J* = 6.8 Hz, 2H). ^13^C-NMR (CDCl_3_) δ 159.7, 156.1, 148.9, 138.9, 136.7, 129.1, 123.6, 123.3, 121.6, 120.7, 39.6, 37.5. HRMS (ESI): *m/z* [M + H]^+^ calcd for C_14_H_16_N_3_O 242.1288, found 242.1285.

1-(2-chlorophenyl)-3-(pyridin-2-ylmethyl)thiourea (ZKT8)

ZKT8 was synthesized from 2-chlorophenyl isothiocyanate (47.2 μL, 0.360 mmol) and 2-picolylamine (35.1 μL, 0.360 mmol) by the same method as ZKT3, and 79.6 mg of white solid was obtained (yield: 79.6%). ^1^H-NMR (CDCl_3_) δ 8.42 (s, 1H), 7.86 (br, 1H), 7.67–7.70 (m, 2H), 7.59 (br, 1H), 7.50 (d, *J* = 8.0 Hz, 1H), 7.36 (t, *J* = 7.5 Hz, 1H), 7.31 (d, *J* = 7.5 Hz, 1H), 7.19–7.26 (m, 2H), 4.93 (s, 2H). ^13^C-NMR (CDCl_3_) δ 180.0, 155.1, 148.6, 136.9, 133.8, 130.5, 129.1, 127.8, 127.6, 126.5, 122.5, 122.1, 49.7. HRMS (ESI): *m/z* [M + H]^+^ calcd for C_13_H_13_ClN_3_S 278.0513, found 278.0504.

1-(4-chlorophenyl)-3-(pyridin-2-ylmethyl)thiourea (ZKT9)

2-Picolylamine (35.1 μL, 0.360 mmol) was added to the solution of 4-chlorophenyl isothiocyanate (61.1 mg, 0.360 mmol) in 2.9 mL acetone, and refluxed for 2 h. The reaction mixture was evaporated and the resulting residue was purified by recrystallization from ethyl acetate and hexane to obtain 88.2 mg white solid (yield: 88.2%). ^1^H-NMR (CDCl_3_) δ 8.43 (s, 1H), 8.06 (br, 1H), 7.85 (br, 1H), 7.68 (t, *J* = 7.5 Hz, 1H), 7.39 (d, *J* = 9.0 Hz, 2H), 7.19–7.30 (m, 4H), 4.91 (s, 2H). ^13^C-NMR (CDCl_3_) δ 179.8, 155.1, 148.7, 136.9, 135.2, 132.0, 129.9, 125.7, 122.6, 122.1, 46.6. HRMS (ESI): *m/z* [M + H]^+^ calcd for C_13_H_13_ClN_3_S 278.0513, found 278.0509.

1-(4-fluorophenyl)-3-(pyridin-2-ylmethyl)thiourea (ZKT10)

ZKT10 was synthesized from 4-fluorophenyl isothiocyanate (46.6 μL, 0.380 mmol) and 2-picolylamine (37.3 μL, 0.380 mmol) by the same method as ZKT9 and 87.3 mg of white solid was obtained (yield: 87.9%). ^1^H-NMR (CDCl_3_) δ 8.41 (s, 1H), 7.81 (br, 1H), 7.68 (t, *J* = 7.0 Hz, 2H), 7.28–7.30 (m, 3H), 7.19 (t, *J* = 6.0 Hz, 1H), 7.12–7.16 (m, 2H), 4.91 (s, 2H). ^13^C-NMR (CDCl_3_) δ 180.2, 161.0 (d, *J* = 245.6 Hz), 155.2, 148.6, 136.8, 132.5, 127.1 (d, *J* = 7.1 Hz), 122.3 (d, *J* = 48.9 Hz), 116.7 (d, *J* = 21.5 Hz), 49.5. HRMS (ESI): *m/z* [M + H]^+^ calcd for C_13_H_13_FN_3_S 262.0809, found 262.0801.

1-(3,4-dichlorophenyl)-3-(pyridin-2-ylmethyl)thiourea (ZKT11)

ZKT11 was synthesized from 3,4-dichlorophenyl isothiocyanate (45.6 μL, 0.320 mmol) and 2-picolylamine (31.2 μL, 0.320 mmol) by the same method as ZKT3 and 84.3 mg of pale green solid was obtained (yield: 84.3%). ^1^H-NMR (CDCl_3_) δ 8.48 (s, 1H), 8.11 (br, 1H), 7.96 (br, 1H), 7.71 (t, *J* = 6.5 Hz, 1H), 7.47–7.53 (m, 2H), 7.18–7.31 (m, 3H), 4.92 (s, 2H). ^13^C-NMR (DMSO-d_6_) δ 180.6, 157.1, 148.8, 139.8, 136.8, 130.5, 130.3, 125.4, 123.7, 122.5, 122.3, 121.5, 48.8. HRMS (ESI): *m/z* [M + H]^+^ calcd for C_13_H_12_Cl_2_N_3_S 312.0123, found 312.0123.

1-(2,4-dichlorophenyl)-3-(pyridin-2-ylmethyl)thiourea (ZKT12)

2-Picolylamine (202 μL, 2.00 mmol) was added to the solution of 2,4-dichlorophenyl isothiocyanate (408 mg, 2.00 mmol) in 23 mL dichloromethane and the reaction mixture was stirred at room temperature for an hour. The precipitated white solid was purified by recrystallization from ethyl acetate and hexane to obtain 520 mg of white needle-like crystals (yield: 83.3%). ^1^H-NMR (CDCl_3_) δ 8.45 (s, 1H), 7.88 (br, 1H), 7.61–7.72 (m, 3H), 7.50 (d, *J* = 2.0 Hz, 1H), 7.30-7.34 (m, 2H), 7.23 (t, *J* = 5.3 Hz, 1H), 4.91 (s, 2H). ^13^C-NMR (DMSO-d_6_) δ 181.9, 157.4, 148.8, 136.7, 135.4, 130.5, 130.4, 130.3, 128.8, 127.3, 122.2, 121.4, 49.2. HRMS (ESI): *m/z* [M + H]^+^ calcd for C_13_H_12_Cl_2_N_3_S 312.0123, found 312.0121.

1-(2,3-dichlorophenyl)-3-(pyridin-2-ylmethyl)thiourea (ZKT13)

ZKT13 was synthesized from 2,3-dichlorophenyl isothiocyanate (45.3 μL, 0.320 mmol) and 2-picolylamine (31.2 μL, 0.320 mmol) by the same method as ZKT3 and 80.6 mg of white solid was obtained (yield: 80.6%). ^1^H-NMR (CDCl_3_) δ 8.43 (s, 1H), 7.99 (br, 1H), 7.69–7.75 (m, 2H), 7.57 (br, 1H), 7.38-7.40 (m, 1H), 7.26–7.31 (m, 2H), 7.22 (s, 1H), 4.93 (br, 2H). ^13^C-NMR (DMSO-d_6_) δ 181.9, 157.5, 148.8, 138.2, 136.7, 131.7, 128.2, 128.2, 127.7, 122.2, 121.4, 49.3. HRMS (ESI): *m/z* [M + H]^+^ calcd for C_13_H_12_Cl_2_N_3_S 312.0123, found 312.0123.

1-(4-ethoxyphenyl)-3-(pyridin-2-ylmethyl)thiourea (ZKT14)

ZKT14 was synthesized from 4-ethoxyphenyl isothiocyanate (59.1 μL, 0.350 mmol) and 2-picolylamine (34.1 μL, 0.350 mmol) by the same method as ZKT3 and 85.9 mg of white solid was obtained (yield: 85.5%). ^1^H-NMR (CDCl_3_) δ 8.41 (d, *J* = 4.5 Hz, 1H), 7.63–7.68 (m, 2H), 7.48 (br, 1H), 7.30 (d, *J* = 7.5 Hz, 1H), 7.16–7.21 (m, 3H), 6.94 (td, *J* = 9.5 Hz, 2.8 Hz, 2H), 4.92 (d, *J* = 2.5 Hz, 2H), 4.05 (q, *J* = 7.0 Hz, 2H), 1.44 (t, *J* = 7.0 Hz, 3H). ^13^C-NMR (CDCl_3_) δ 180.5, 158.0, 155.5, 148.7, 136.7, 128.6, 127.2, 122.4, 122.0, 115.6, 63.7, 49.8, 14.7. HRMS (ESI): *m/z* [M + H]^+^ calcd for C_15_H_18_N_3_OS 288.1165, found 288.1173.

1-(4-*tert*-butylphenyl)-3-(pyridin-2-ylmethyl)thiourea (ZKT15)

ZKT15 was synthesized from 4-*tert*-butylphenyl isothiocyanate (63.1 mg, 0.330 mmol) and 2-picolylamine (32.2 μL, 0.330 mmol) by the same method as ZKT2 and 76.5 mg of white solid was obtained (yield: 77.4%). ^1^H-NMR (CDCl_3_) δ 8.42 (d, *J* = 4.5 Hz, 1H), 7.64–7.72 (m, 3H), 7.45 (d, *J* = 8.5 Hz, 2H), 7.32 (d, *J* = 7.5 Hz, 1H), 7.17–7.22 (m, 3H), 4.94 (s, 2H), 1.34 (s, 9H). ^13^C-NMR (CDCl_3_) δ 180.1, 155.6, 150.0, 148.7, 136.8, 133.6, 126.8, 124.4, 122.4, 122.1, 49.8, 34.6, 31.2. HRMS (ESI): *m/z* [M + H]^+^ calcd for C_17_H_22_N_3_S 300.1529, found 300.1521.

1,3-diphenylthiourea (ZKT16)

Aniline (40.1 μL, 0.440 mmol) was added to the solution of phenyl isothiocyanate (52.6 μL, 0.440 mmol) in 3.3 mL of acetone, and refluxed for 4 h. The reaction mixture was evaporated and the resulting residue was purified by recrystallization from ethanol and acetone to obtain 50.3 mg of white solid (yield: 50.1%). ^1^H-NMR (CDCl_3_) δ 7.81 (s, 2H), 7.38–7.44 (m, 8H), 7.28–7.31 (m, 2H). ^13^C-NMR (CDCl_3_) δ 179.6, 137.0, 129.5, 126.9, 125.2. HRMS (ESI): *m/z* [M + H]^+^ calcd for C_13_H_13_N_2_S 229.0794, found 229.0792.

1-benzyl-3-phenylthiourea (ZKT17)

Triethylamine (61.0 μL, 0.440 mmol) and benzyl isothiocyanate (64.0 μL, 0.504 mmol) were added to the solution of aniline (38.3 μL, 0.420 mmol) in 1.7 mL *N,N*-dimethylformamide and the reaction mixture was stirred at 60 °C overnight. Ethyl acetate was added to the reaction mixture and it was washed with saturated NaCl. The organic layer was dried by sodium sulfate and the solvent was removed under reduced pressure. The resulting residue was purified by recrystallization from ethanol and hexane to obtain 78.2 mg of white solid (yield: 78.8%). ^1^H-NMR (CDCl_3_) δ 7.78 (br, 1H), 7.39-7.43 (m, 2H), 7.26–7.35 (m, 6H), 4.20–7.22 (m, 2H), 6.25 (br, 1H), 4.89 (d, *J* = 4.0 Hz, 2H). ^13^C-NMR (CDCl_3_) δ 180.7, 137.2, 135.9, 130.2, 128.7, 127.7, 127.5, 127.3, 125.2, 49.3. HRMS (ESI): *m/z* [M + H]^+^ calcd for C_14_H_15_N_2_S 243.0950, found 243.0948.

1-phenetyl-3-phenylthiourea (ZKT18)

Phenyl isothiocyanate (120 μL, 1.01 mmol) was added to the solution of phenethylamine (150 μL, 1.19 mmol) in 11.5 mL dichloromethane and was stirred for an hour. The reaction mixture was passed through a short silica gel column eluted by ethyl acetate and the solvent was removed under reduced pressure. The resulting residue was purified by silica gel column chromatography (hexane:ethyl acetate = 4:1) to obtain 258 mg of white solid (yield: 100%). ^1^H-NMR (CDCl_3_) δ 7.89 (s, 1H), 7.20-7.33 (m, 6H), 7.13–7.14 (m, 2H), 6.99–7.00 (m, 2H), 5.98 (s, 1H), 3.89 (q, *J* = 6.2 Hz, 2H), 2.91 (t, *J* = 6.8 Hz, 2H). ^13^C-NMR (CDCl_3_) δ 180.4, 138.4, 135.7, 130.1, 128.7, 127.2, 126.6, 125.1, 46.3, 34.7. HRMS (ESI): *m/z* [M + H]^+^ calcd for C_15_H_17_N_2_S 257.1107, found 257.1109.

### 4.3. Plant Materials and Growth Conditions

Seeds of *Arabidopsis thaliana* (ecotype Columbia *ein2-5* and *ein3 eil1*) were supplied by Dr. K. Jiang of the Southern University of Science and Technology. For experiments with etiolated seedlings, the seeds were sterilized in 70% ethanol for 20 min and washed with 99% ethanol. Seeds were sown on 0.8% solidified agar medium containing 1/2 Murashige and Skoog (MS) salt, 0.1% 1000× vitamin stock, 1% sucrose, and the compounds, were added to 6-well plates. The plates were wrapped with aluminum foil and pre-incubated at 4 °C for 3 d. After pre-incubation, the aluminum foil was removed, and the plates were inserted into a growth chamber at 22 °C for 3 h. The plates were then wrapped with two layers of aluminum foil, and the plants were grown in a growth chamber at 22 °C for 77–80 h. The seedlings were transferred to an OHP film, and images were obtained using a scanner (EPSON GT-X820). The effect of the test compounds on the growth of *Arabidopsis* seedlings was determined by measuring the hypocotyl length and the angle of the apical hook using ImageJ 1.48v. For real-time quantitative reverse transcription polymerase chain reaction (qRT-PCR) analysis, seeds were sterilized with 1 mL of 1% sodium hypochlorite with a drop of Tween 20 for 20 min and washed fivefold with sterile distilled water. The seeds were soaked in sterile distilled water and kept at 4 °C for 2 d. The seeds were sown on 2.5 mL 1/2 MS liquid medium added to 12-well plates and were grown under illuminated conditions at 22 °C for 7 d. After 7 d, a stock solution of the compounds was added to the medium, and the plants were sampled 3 and 24 h after treatment. Stock solutions of all chemicals, except ACC and AIBA, were dissolved in DMSO and stocked at –30 °C before use. Stock solutions of ACC and AIBA were prepared in sterile distilled water. DMSO was added below 0.1% (*v/v*) to the growth medium in all experiments.

### 4.4. Isolation of Total RNA

The samples were frozen in liquid nitrogen and crushed in a hard master tube (2.0 mL) (Biomedical Science Co., Ltd., Tokyo, Japan) with zirconia beads using a Shake Master NEO (Biomedical Science Co., Ltd.). Total RNA was extracted from tissues using a Total RNA Extraction Kit Mini (Plant) (RBC bioscience Corp., New Taipei, Taiwan). The isolated total RNA samples were stored at –80 °C before use.

### 4.5. qRT-PCR Analysis

cDNAs were synthesized from 100 ng of total RNA using ReverTra Ace qPCR RT Master Mix with gDNA Remover (TOYOBO Co., Ltd., Osaka, Japan). qRT-PCR was performed on a Thermal Cycler Dice Real Time System TP800 (Takara Bio Co., Ltd., Shiga, Japan) with primers listed in Table 3 using a KAPA SYBR FAST 1PCR Kit (Nippon Genetics Co., Ltd., Tokyo, Japan).

### 4.6. Statistical Analysis

Data analyses (*t*-test and analysis of variance) were performed to determine the significant difference using GraphPad Prism 9 (GraphPad Software, Boston, MA, USA). Dose-response curves and EC_50_ values were also calculated using GraphPad Prism 9 with a four-parameter logistic curve model.

## 5. Conclusions

In conclusion, our study revealed that thiourea derivatives containing pyridinyl or phenyl moieties exhibited ethylene-like activity by affecting the upstream components of the ethylene signaling pathway. The mechanism of ethylene signaling activation is hypothesized to involve binding to the copper cofactor of ethylene receptors or influencing copper transport to ethylene receptors. Among the ZKT1 derivatives tested, ZKT2, 3, 4, 11, and 18 showed the highest biological activity in inducing the “triple response” in *Arabidopsis*. These ZKT1 derivatives have the potential to serve as potent ethylene mimics for controlling ethylene response in plants.

## Figures and Tables

**Figure 1 ijms-24-12420-f001:**
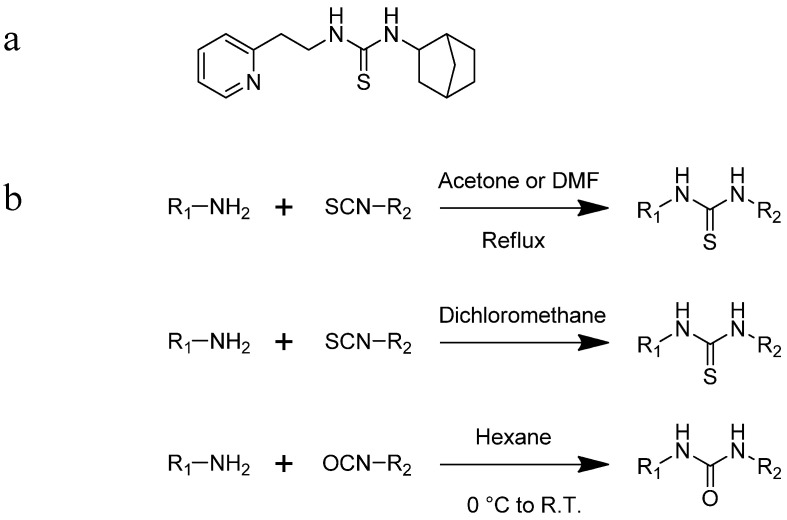
(**a**) Structure of a selected compound with ethylene-like activity (ZKT1). (**b**) General methods for the synthesis of ZKT1 derivatives. Primary amine and isothiocyanate were refluxed in acetone or DMF or mixed in dichloromethane to obtain ZKT1 derivatives. A urea derivative ZKT7 was synthesized from primary amine and isocyanate by mixing both compounds in hexane. R.T. stands for room temperature.

**Figure 2 ijms-24-12420-f002:**
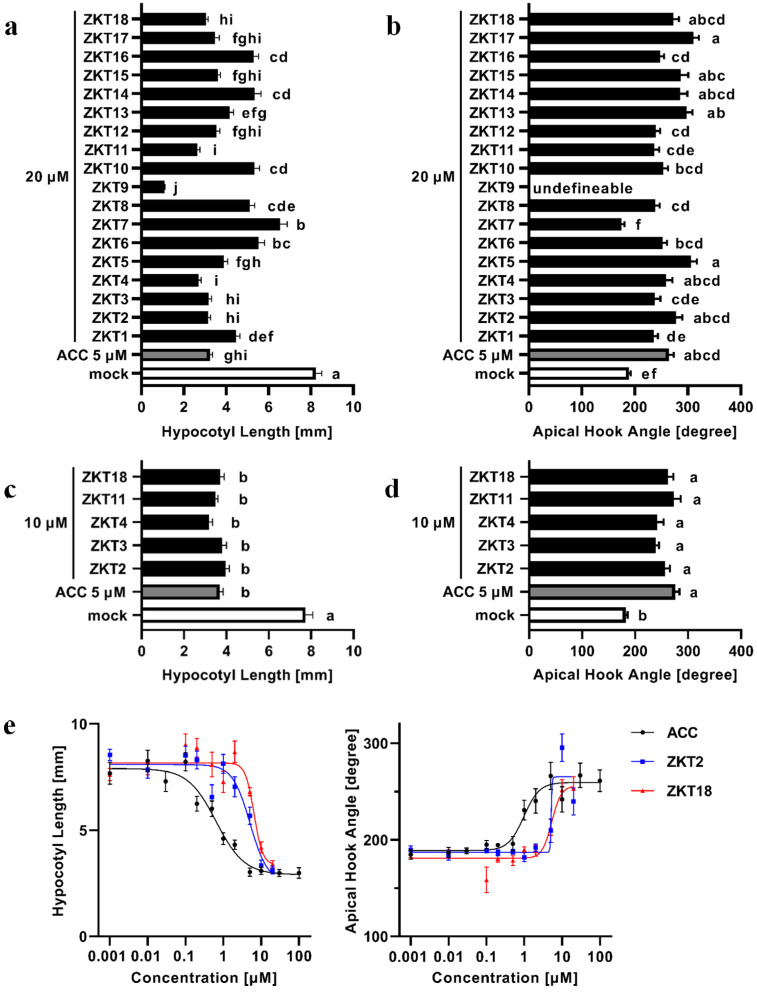
Morphological characters of Col-0 seedlings grown on 0.8% agar solidified 1/2 MS medium with compounds under 22 °C dark conditions for 3 d. (**a**) Hypocotyl length of seedlings treated with 5 μM of ACC or 20 μM of ZKT1 derivatives. Bars are means ± S.E., *n* ≥ 32. Bars with different letters are significantly different at *p* < 0.05 by one-way ANOVA followed by Tukey’s HSD test. (**b**) The apical hook angle of seedlings treated with derivatives of 5 μM of ACC or 20 μM of ZKT1 derivatives. The apical hook angle of seedlings treated with ZKT9 was undefinable due to cotyledon opening. Bars are means ± S.E., *n* ≥ 1. Bars with different letters are significantly different at *p* < 0.05 by one-way ANOVA, followed by Tukey’s HSD test. (**c**) Hypocotyl length of seedlings treated with derivatives of 5 μM of ACC or 10 μM of ZKT1 derivatives. Bars are means ± S.E., *n* ≥ 34. Bars with different letters are significantly different at *p* < 0.05 by one-way ANOVA, followed by Tukey’s HSD test. (**d**) The apical hook angle of seedlings treated with derivatives of 5 μM of ACC or 10 μM of ZKT1 derivatives. Bars are means ± S.E., *n* ≥ 34. Bars with different letters are significantly different at *p* < 0.05 by one-way ANOVA, followed by Tukey’s HSD test. (**e**) Dose-response curves of hypocotyl length and apical hook angle of seedlings treated with 0.001–100 μM of ACC or 0.001–20 μM of ZKT2 or ZKT18. Points are means ± S.E., *n* ≥ 13. Dose-response curves were drawn as four-parameter logistic curves using GraphPad Prism 9.

**Figure 3 ijms-24-12420-f003:**
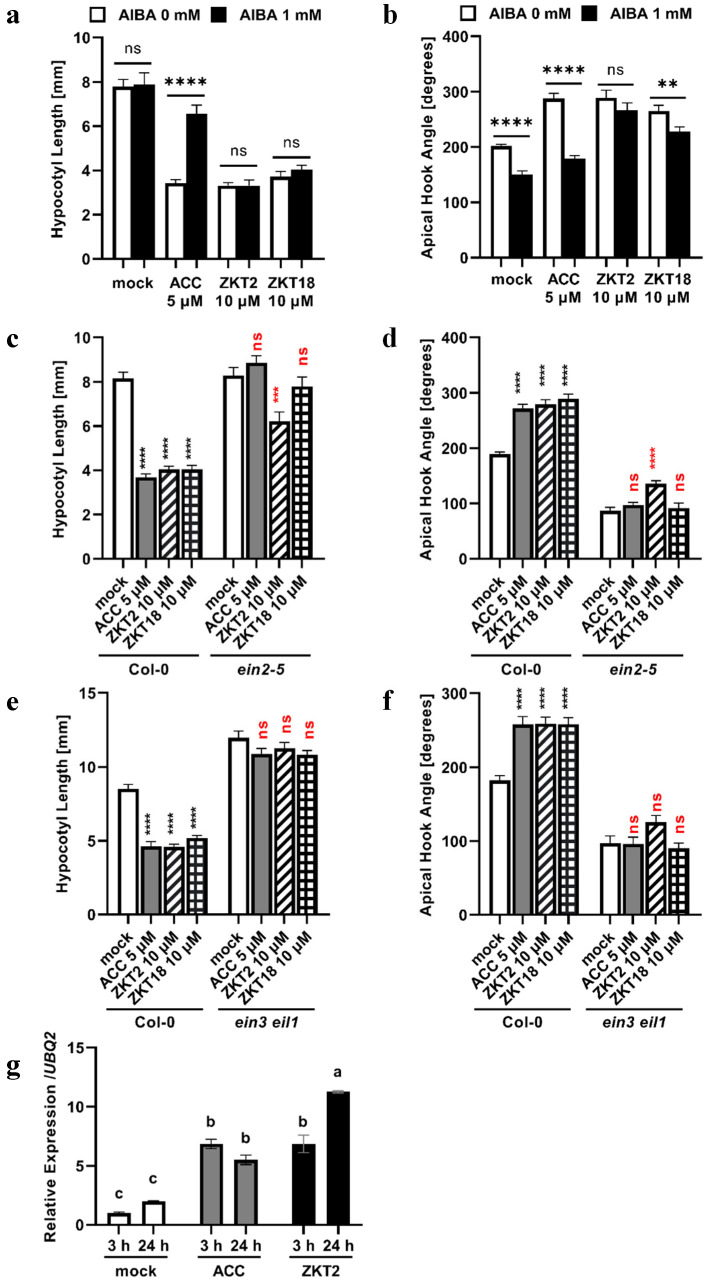
(**a**–**f**) Morphological characters of *A. thaliana* seedlings grown on 0.8% agar solidified 1/2 MS medium with compounds under 22 °C dark conditions for 3 d. Bars are means ± S.E. (**a**) Hypocotyl length of Col-0 seedlings treated with 5 μM of ACC or 10 μM of ZKT2 or ZKT18 with or without 1 mM of AIBA. *n* ≥ 26, ns: not significantly different, ****: *p* < 0.0001, *t*-test. (**b**) The apical hook angle of Col-0 seedlings treated with 5 μM of ACC or 10 μM of ZKT2 or ZKT18 with or without 1 mM of AIBA. *n* ≥ 26, ns: not significantly different, **: *p* < 0.01, ****: *p* < 0.0001, *t*-test. (**c**) Hypocotyl length of Col-0 and *ein2-5* seedlings treated with 5 μM of ACC or 10 μM of ZKT2 or ZKT18. *n* ≥ 27, ns: not significantly different, ***: *p* < 0.001, ****: *p* < 0.0001, one-way ANOVA followed by Dunnett’s test vs. Col-0 mock (black) or *ein2-5* mock (red). (**d**) The apical hook angle of Col-0 and *ein2-5* seedlings treated with 5 μM of ACC or 10 μM of ZKT2 or ZKT18. *n* ≥ 27, ns: not significantly different, ****: *p* < 0.0001, one-way ANOVA followed by Dunnett’s test vs. Col-0 mock (black) or *ein2-5* mock (red). (**e**) Hypocotyl length of Col-0 or *ein3 eil1* seedlings treated with 5 μM of ACC or 10 μM of ZKT2 or ZKT18. *n* ≥ 32, ns: not significantly different, ****: *p* < 0.0001, one-way ANOVA followed by Dunnett’s test vs. Col-0 mock (black) or *ein3 eil1* mock (red). (**f**) The apical hook angle of Col-0 or *ein3 eil1* seedlings treated with 5 μM of ACC or 10 μM of ZKT2 or ZKT18. *n* ≥ 32, ns: not significantly different, ****: *p* < 0.0001, one-way ANOVA followed by Dunnett’s test vs. Col-0 mock (black) or *ein3 eil1* mock (red). (**g**) Col-0 seedlings were grown on 1/2 MS liquid medium without compounds under 22 °C illuminated conditions for 7 d and treated with 10 μM of ACC or 30 μM of ZKT2. Relative expression levels of the *ERF1* gene were measured 3 and 24 h after treatment. Bars are means ± S.E., *n* = 4. Bars with different letters are significantly different at *p* < 0.05 by one-way ANOVA, followed by Tukey’s HSD test.

**Table 1 ijms-24-12420-t001:** List of the 17 synthesized ZKT1 derivatives.

Entry	Structure	Entry	Structure
ZKT2	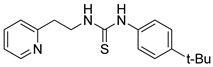	ZKT11	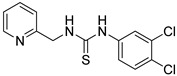
ZKT3	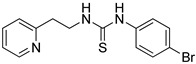	ZKT12	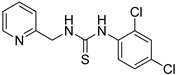
ZKT4	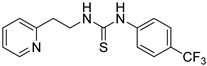	ZKT13	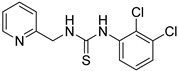
ZKT5	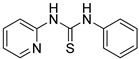	ZKT14	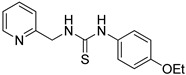
ZKT6	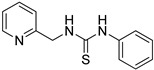	ZKT15	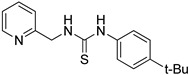
ZKT7	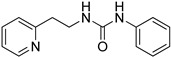	ZKT16	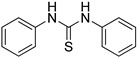
ZKT8	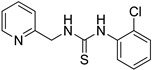	ZKT17	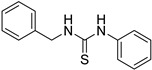
ZKT9	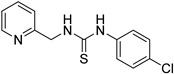	ZKT18	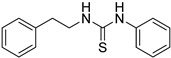
ZKT10	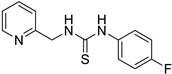		

**Table 2 ijms-24-12420-t002:** EC_50_ values of ACC, ZKT2, and ZKT18 were calculated from the hypocotyl length or apical hook angle. These values were calculated using GraphPad Prism 9.0.

Entry	EC_50_ Values Calculated from Hypocotyl Length	EC_50_ Values Calculated from Apical Hook Angle
ACC	0.648 (μM)	0.976 (μM)
ZKT2	5.07 (μM)	5.14 (μM)
ZKT18	6.49 (μM)	5.31 (μM)

**Table 3 ijms-24-12420-t003:** Primers for qRT-PCR.

Primer Name	Forward/Reverse	Primer Sequence
AtERF1F_Fwd	F	AGACGACGGCCATGGGGGAA
AtETR1R_Rev	R	TCCGCGCTTTCGAACGTCCC
AtUBQ2_Fwd	F	CCAAGATCCAGGACAAAGAAGGA
AtUBQ2_Rev	R	TGGAGACGAGAGCATAACACTTGC

## Data Availability

Not applicable.

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
