# Peer review of "Small Molecules with Thiourea Skeleton Induce Ethylene Response in Arabidopsis"

_ijms, 2023, doi:10.3390/ijms241512420_

Round 1
Reviewer 1 Report
The publication "Small molecules with thiourea skeleton induce ethylene response in Arabidopsis" concerns the synthesis of aromatic and heterocyclic derivatives of thiourea and their use as compounds with biological activity.
The work is interesting, but I have a few comments.
1) The Figure 1 is missing.
2) For the spectrum 1H of the described compounds: ZT9, ZT12, ZT13, ZT15, ZT19 1 hydrogen atom is missing. In turn, for the ZT10 compound there is an additional 1 hydrogen atom in spectral data. In the description of the spectrum (line 451) there is no entry, I suppose 1H.
Please assign the appropriate hydrogen atoms to their structures. 3) For the 13C spectra of ZT19 and ZT20, 1 carbon atom signal is missing in the spectral data description. For the ZT40 derivative, as many as 3 signals are missing from the data description. In addition, there was a signal 30.09 (line 493), it should probably be 130.09 ppm. 4) The description of the ZT13 mass spectrum does not correspond to the real formula (presumably the formula was rewritten as ZT12).5) Conclusions must also be added by the Authors of the publication.
6) The numbering in Figure 3 does not quite match the text (e.g. Figure 3 G, line 182).
7) In some places of the text there is a different font.
Author Response
Responses to the comments from Reviewer 1
Due to a compound duplication error, the compound number was changed in the revised manuscript. To avoid confusion the name of the compound was changed from ZT to ZKT according to the comment from Reviewer 3. To make the renaming of compounds easier to understand, we have compiled a table with the name of each compound after it has been changed. Please see the attachment.
1) The Figure 1 is missing.
(response)
We added the Figure 1 (P2, L76).
2) For the spectrum 1H of the described compounds: ZT9, ZT12, ZT13, ZT15, ZT19 1 hydrogen atom is missing. In turn, for the ZT10 compound there is an additional 1 hydrogen atom in spectral data. In the description of the spectrum (line 451) there is no entry, I suppose 1H. Please assign the appropriate hydrogen atoms to their structures.
(response)
We carefully corrected the NMR data for the compounds you mentioned as well as other compounds. (P11, L354, L364, L374, L398, P12, L416 and P13, L461 and L462).
3) For the 13C spectra of ZT19 and ZT20, 1 carbon atom signal is missing in the spectral data description. For the ZT40 derivative, as many as 3 signals are missing from the data description. In addition,there was a signal 30.09 (line 493), it should probably be 130.09 ppm.
(response)
We carefully corrected the spectra information (P12, L418 and P13, L504). For ZKT11(ZT20), the signal at 148.61 ppm represents two carbons. For ZKT19(ZT40), the signal at 128.71 ppm peak represents two carbons.
4) The description of the ZT13 mass spectrum does not correspond to the real formula (presumably the formula was rewritten as ZT12).
(response)
We corrected the molecular formula (P11, L377).
5) Conclusions must also be added by the Authors of the publication.
(response)
Thank you for your advice. We added the conclusion (P10, L310-319).
6) The numbering in Figure 3 does not quite match the text (e.g. Figure 3 G, line 182).
(response)
We corrected the numbering in Figure 3 (P6, L158, L169, L178, L186, P7, L194-P8, L214).
7) In some places of the text there is a different font.
(response)
We corrected so that the font used in this manuscript are identical (P15, L552-P17, L644).

Reviewer 2 Report
The manuscript represents the developement of solid ethylene response inducers. The authors synthesized several thiourea (and one urea) derivatives, which were characterized, and their structure-activity relationships were studied, and also the mechanism of their action. My suggestions for corrections are the followings:
Figure 1 is missing
Tables 1 and 2 should not be images. Table 2 should be in a normal (word format) table according to the template. In the case of Table 1, the authors can use a Figure (image) with structures and names, but without a table format, or a word format table with images of each structures (such as e.g. Table 2 in Molecules 2020, 25(13), 3011).
Line 114: "these five ZT compounds": do the authors mean six compounds or what five of the six? Because there are six ZT compounds listed in the previous sentence and in Figure 2C, D.
Captions of Figures 2A, B: "with derivatives of 5 μM of ACC or 20 μM of ZT1" → "with 5 μM of ACC or 20 μM of derivatives of ZT1"
Captions of Figures 2C, D: "with derivatives of 5 μM of ACC or 10 μM of ZT1" → "with 5 μM of ACC or 10 μM of derivatives of ZT1"
Numbering of subchapters in "3. Discussion" is missing
Line 214: "1-Phenyl" → "1-phenyl"
Line 224: "1-phenety" → "1-phenetyl"
The paragraph between lines 294 and 301 should be in "5. Conclusions" chapter
In "4. Materials and Methods":
In the cases of ZT9, 12, 13, 15, 19, the sum of the listed protons (1H+2H+… etc.) is one less than in the molecules. The authors should check their spectra (for comparison, ZT13: Tetrahedron 56 (2000) 629–637, ZT19: Tetrahedron Letters 55 (2014) 6769–6772, suppl).
In the case of ZT10, the sum of the listed protons is one more than in the molecule.
In the case of ZT24, the proton number for the first 1H NMR signal is missing (please, then check the sum of the listed protons also).
In the case of ZT19, the aliphatic carbon signal is missing.
In the case of ZT40, the two aliphatic carbon signals are missing, and maybe 30.09 should be 130.09 (for comparison, ZT40: Tetrahedron 56 (2000) 629–637).
Line 491: the eluent for column chromatography is missing
Lines 324 and 374 (for ZT8 and ZT14): "The reaction mixture was evaporated and passed through short silica-gel column." Did the authors interchange the two procedures in the latter sentence? (For ZT40 they wrote: "The reaction mixture was passed through short silica gel column and the solvent was removed under reduced pressure.")
Lines 321 and 457: "(4-(tert-butyl)phenyl)" → "(4-tert-butylphenyl)"
Lines 325, 375, 490: "silica-gel" → "silica gel"
Lines 362 and 371: "(phenyl)" → "phenyl"
Lines 369 and 380: the authors should use the "HRMS (ESI): m/z [M+H]+ calcd for …, found …" format, similarly to the other cases
Line 369: the molecular formula for ZT13 is not correct
Line 382: "pheny" → "phenyl"
Lines 403, 414, 423, 431, 442, 450, 459: "2-pycolylamine" → "2-picolylamine"
English language is fine. I noticed some missing indefinite articles (before: "costly equipment", and "short silica gel column" 3 times)
Author Response
Responses to the comments from Reviewer 2
Due to a compound duplication error, the compound number was changed in the revised manuscript. To avoid confusion the name of the compound was changed from ZT to ZKT according to the comment from Reviewer 3. To make the renaming of compounds easier to understand, we have compiled a table with the name of each compound after it has been changed. Please see the attachement.
1) Figure 1 is missing
(response)
We added Figure 1 (P2, L76).
2) Tables 1 and 2 should not be images. Table 2 should be in a normal (word format) table according to the template. In the case of Table 1, the authors can use a Figure (image) with structures and names, but without a table format, or a word format table with images of each structures (such as e.g. Table 2 in Molecules 2020, 25(13), 3011).
(response)
According to the comments, we applied word format to tables (P3, L88 and P4, L130).
3) Line 114: "these five ZT compounds": do the authors mean six compounds or what five of the six? Because there are six ZT compounds listed in the previous sentence and in Figure 2C, D.
(response)
We noticed that ZT9(ZKT3) and ZT10 were identical and thus deleted ZT10 from the manuscript. Therefore, "five" in the original version was wrong but now "five" in the revised version is correct.
4) Captions of Figures 2A, B: "with derivatives of 5 μM of ACC or 20 μM of ZT1" → "with 5 μM of ACC or 20 μM of derivatives of ZT1" Captions of Figures 2C, D: "with derivatives of 5 μM of ACC or 10 μM of ZT1" → "with 5 μM of ACC or 10 μM of derivatives of ZT1"
(response)
According to the comments, we corrected the sentences (P5, L137, L139, L143 and L145).
5) Numbering of subchapters in "3. Discussion" is missing
(response)
We added numbering of subchapters in the discussion part ((P8, L217, L251, L292).
6) Line 214: "1-Phenyl" → "1-phenyl"
(response)
We changed the capital letter to the small letter (P8, L226).
7) Line 224: "1-phenety" → "1-phenetyl"
(response)
We corrected the spelling (P8, L236).
8) The paragraph between lines 294 and 301 should be in "5. Conclusions" chapter.
(response)
We moved the paragraph to the conclusion chapter (P10, L310-L318).
In "4. Materials and Methods":
9) In the cases of ZT9, 12, 13, 15, 19, the sum of the listed protons (1H+2H+… etc.) is one less than in the molecules. The authors should check their spectra (for comparison, ZT13: Tetrahedron 56 (2000) 629–637, ZT19: Tetrahedron Letters 55 (2014) 6769–6772, suppl).
(response)
We corrected the number of protons (P11, L354, L362, L374, L398, P12, L416).
10) In the case of ZT10, the sum of the listed protons is one more than in the molecule.
(response)
We found that ZT9 (ZKT3) and ZT10 were identical and thus deleted ZT10 from the manuscript.
The number of protons was corrected.
11) In the case of ZT24, the proton number for the first 1H NMR signal is missing (please, then check the sum of the listed protons also).
(response)
In the revised version, ZT24 was renamed as ZKT14.
We corrected the number of protons (P13, L461 and L462).
12) In the case of ZT19, the aliphatic carbon signal is missing.
(response)
We corrected the spectra information (P12, L418) of ZT19 (ZKT9 in the new version).
13) In the case of ZT40, the two aliphatic carbon signals are missing, and maybe 30.09 should be 130.09 (for comparison, ZT40: Tetrahedron 56 (2000) 629–637).
(response)
We corrected the spectra information (P13, L504) of ZT40 (ZKT18 in the new version).
14) Line 491: the eluent for column chromatography is missing.
(response)
We added the eluent information (P13, L501)
15) Lines 324 and 374 (for ZT8 and ZT14): "The reaction mixture was evaporated and passed through short silica-gel column." Did the authors interchange the two procedures in the latter sentence? (ForZT40 they wrote: "The reaction mixture was passed through short silicagel column and the solvent was removed under reduced pressure.").
(response)
We used different methods in purifying the former ones and the latter one.
16) Lines 321 and 457: "(4-(tert-butyl)phenyl)" → "(4-tert-butylphenyl)"
(response)
We corrected the spelling (P10, L338 and P12, L467).
17) Lines 325, 375, 490: "silica-gel" → "silica gel"
(response)
We corrected the spelling (P10, L342, P11, L383, P13, L499 and L501).
18) Lines 362 and 371: "(phenyl)" → "phenyl"
(response)
We corrected the spelling (P11, L370 and L379).
19) Lines 369 and 380: the authors should use the "HRMS (ESI): m/z [M+H]+calcd for …, found …" format, similarly to the other cases.
(response)
We corrected the sentences (P11, L377 and L388).
20) Line 369: the molecular formula for ZT13 is not correct.
(response)
We corrected the molecular formula (P11, L377) of ZT13 (ZKT5 in the new version).
21) Line 382: "pheny" → "phenyl"
(response)
We corrected the spelling (P11, L391).
22) Lines 403, 414, 423, 431, 442, 450, 459: "2-pycolylamine" → "2-picolylamine"
(response)
We corrected the spellings (P12, L412, L423, L432, L440, L451, L460 and P13, L469).

Reviewer 3 Report
This paper describes the synthesis and the screening of a library of thiourea derivative having ethylene-mimic behaviour to control the Arabidopsis thaliana. Authors
- pag. 2 lines 69-74 the authors write that they screened several compounds, but did not indicate how many or what type of compounds were used to then select the thiourea derivative which is not shown; Are the compounds shown in in figure 1? The lack of Figure 1 does not allow to follow the synthesis of the parent compound for the creation of the library.
- Table 1: compounds must be numbered consecutively; the fact that the authors synthesized compounds that are not shown is irrelevant to the article
- pag. 4 line 122 the authors indicate a picture in figure 2 (fig 2E) that does not exist.
- in figure 2 A, B, C and D it is not clear what the letters on the bars refer to
- pag. 6 line 154, line 165, line 173, line 183: the statements in the text refer to wrong parts of figure 3 or to non-existent parts.
It is difficult to follow the description and the conclusions, compared to the experimentation, due to the inaccuracies in referring to the data or their lack. Furthermore, the purity and nature of the synthesized compounds is uncertain as the acquired NMR spectra are totally missing.
I believe that in this form the manuscript should be rejected.
I regret with the authors for the little care they took in presenting their data and in checking the accuracy and presence of all parts in the version sent for refereeing, wasting their time and that of the referees who worked hard in trying to understand what the authors meant in their manuscript.
Author Response
Responses to the comments from Reviewer 3
Due to a compound duplication error, the compound number was changed in the revised manuscript. To avoid confusion the name of the compound was changed from ZT to ZKT according to the comment from Reviewer 3. To make the renaming of compounds easier to understand, we have compiled a table with the name of each compound after it has been changed. Please see the attachement.
1) pag. 2 lines 69-74 the authors write that they screened several compounds, but did not indicate how many or what type of compounds were used to then select the thiourea derivative which is not shown; Are the compounds shown in in figure 1? The lack of Figure 1 does not allow to follow the synthesis of the parent compound for the creation of the library.
(response)
We added the information of the library we screened and Figure 1 (P2,
L69-70 and L76).
2) Table 1: compounds must be numbered consecutively; the fact that the authors synthesized compounds that are not shown is irrelevant to the article
(response)
We renamed the compounds consecutively (P3, L88).
3) pag. 4 line 122 the authors indicate a picture in figure 2 (fig 2E)
that does not exist.
(response)
We corrected the numbering in Figure 2 (P4, L100, L116, L124 and P5, L135-P6, L150).
4) in figure 2 A, B, C and D it is not clear what the letters on the
bars refer to.
(response)
We compared the biological activity of chemicals to each other. Bars with different letters are significantly different at p<0.05.
5) pag. 6 line 154, line 165, line 173, line 183: the statements in the text refer to wrong parts of figure 3 or to non-existent parts.
(response)
We corrected the numbering in Figure 3 (P6, L158, L169, L178, L186, P7,
L194-P8, L214).
6) It is difficult to follow the description and the conclusions, compared to the experimentation, due to the inaccuracies in referring to the data or their lack. Furthermore, the purity and nature of the synthesized compounds is uncertain as the acquired NMR spectra are totally missing.
(response)
We would accept your comment. Then, we corrected the numbering in Figures and added the NMR spectra as a supporting information.

Reviewer 4 Report
This manuscript describes structure–activity relationships and investigation on the mode of action of thiourea compounds. It is well written and organized. In addition, the created compound could be promising. Therefore, I think this manuscript could be acceptable as an original paper in IJMS after considering the following comments.
The authors synthesized thioure compounds which have a substituent at 4-position of benzene ring. Why did the authors commit to place the substituent at 4-position? I hope to describe the reason, if possible.
The “50” of EC50 should be written in subscript style throughout the manuscript.
Line 222, “P” in 1-Phenyl is unnecessary to be written in uppercase.
The solvent composition of short silica-gel column should be indicated.
It would be ideal to prepare the spectra of 1H and 13C NMR as the Supporting Information.
Nothing special.
Author Response
Responses to the comments from Reviewer 4
Due to a compound duplication error, the compound number was changed in the revised manuscript. To avoid confusion the name of the compound was changed from ZT to ZKT according to the comment from Reviewer 3. To make the renaming of compounds easier to understand, we have compiled a table with the name of each compound after it has been changed. Please see the attachement.
1) The “50” of EC50 should be written in subscript style throughout the
manuscript.
(response)
We corrected the "50"s of EC50 to subscript style (P4, L130 and P8, L248).
2) Line 222, “P” in 1-Phenyl is unnecessary to be written in uppercase.
(response)
We corrected the spelling (P8, L226).
3) The solvent composition of short silica-gel column should be indicated.
(response)
We added the eluent information (P10, L342, P11, L383 and P13, L499).
4) It would be ideal to prepare the spectra of 1H and 13C NMR as the Supporting Information.
(response)
We added NMR spectra as a supporting information.

Round 2
Reviewer 1 Report
Thank you for submitting the revised version of the manuscript.
I have only minor comments.
1) In the description of Table 1 (line 88) there is: “List of the 18 synthesized ZKT1 derivatives”. There are 17 derivatives of ZKT1.
2) In Figure 1, I suggest additionally insert: hexane (according to the procedure for the compound ZKT7, line 393).
3) The font for the selected words is different (e.g. lines 16-18, 57, 63, 83, 96, 101, 109-111, 118, 126,127, ...).
Author Response
I would appreciate your comments on our manuscript. Followings are our response to your comments.
1) In the description of Table 1 (line 88) there is: “List of the 18
synthesized ZKT1 derivatives”. There are 17 derivatives of ZKT1.
(response)According to your suggestion, we corrected the sentence.
2) In Figure 1, I suggest additionally insert: hexane (according to
the procedure for the compound ZKT7, line 393).
(response) According to your suggestion, we added the synthesis method of ZKT7 in Figure 1.
3) The font for the selected words is different (e.g. lines 16-18, 57,
63, 83, 96, 101, 109-111, 118, 126,127, ...).
(response)According to your suggestion, we standardized the font of the manuscript.
Reviewer 2 Report
The manuscript has been corrected according to the comments, and it became suitable for publication. I suggest only a few very small corrections:
Line 78: "Structure of selected compounds" → "Structure of a selected compound"
Caption of Table 1: "18" → "17"
Table 1: the relative (%) sizes of the structure images should be the same (practically the same as for ZKT2) to provide the same size for e.g. the benzene/pyridine rings, similarly to Table 1 in the original manuscript
Line 148: "µ" → "µM"
Lines 168, 170, 177, 261: "the" should not be in italics
Line 302: "Fig. 2 E" → "Fig. 2 e"
Line 501: "Hexane : Ethyl acetate" → "hexane : ethyl acetate"
English language is fine.
Author Response
I would appreciate your comments on our manuscript. Followings are our responses to your comments.
1) Line 78: "Structure of selected compounds" → "Structure of a selected compound"
(response) According to your suggestion, we corrected the sentence.
2) Caption of Table 1: "18" → "17"
(response) According to your suggestion, we corrected the sentence.
3) Table 1: the relative (%) sizes of the structure images should be the
same (practically the same as for ZKT2) to provide the same size for
e.g. the benzene/pyridine rings, similarly to Table 1 in the original
manuscript
(response) According to your suggestion, we unified the sizes of the structure images.
4) Line 148: "µ" → "µM"
(response) We corrected the spelling.
5) Lines 168, 170, 177, 261: "the" should not be in italics
(response ) We corrected them into upright style.
6) Line 302: "Fig. 2 E" → "Fig. 2 e"
(response) We corrected it.
7) Line 501: "Hexane : Ethyl acetate" → "hexane : ethyl acetate"
(response) We corrected it.
Reviewer 3 Report
The authors have significantly improved the quality of the article and now, with minor adjustments, it can be accepted.
In the experimental part the authors should indicate the chemical shifts of the carbons with only one decimal place, in fact, the reproducibility is lower than for the proton spectra. In the compound ZKT4 the quaternary carbon, 4, directly linked to the CF3 and the signal of the two carbons 3,5 are quartets, therefore the coupling constants C-F must be indicated.
Author Response
I would appreciate your comments on our manuscript. Followings are our responses to your comments.
1) In the experimental part the authors should indicate the chemical
shifts of the carbons with only one decimal place, in fact, the
reproducibility is lower than for the proton spectra.
(reply) We deleted the last digits of all the chemical shifts of the carbons,
thus they are indicated with only one decimal place.
2) In the compound ZKT4 the quaternary carbon, 4, directly linked to the
CF3 and the signal of the two carbons 3,5 are quartets, therefore the
coupling constants C-F must be indicated.
(reply) In order to measure the 13C NMR spectra of ZKT4, we used more than 20
mg of ZKT4 and the scans were more than 23000. However, we could not
observe the 2 quartets supposed to be observed in ZKT4. We do not know the reason why, but this is the fact.